# How Warmer and Drier Conditions Drive Forest Dieback and Tree Death: A Review and Conceptual Model for Silver Fir

**DOI:** 10.3390/plants14213308

**Published:** 2025-10-29

**Authors:** Eustaquio Gil-Pelegrín, José Javier Peguero-Pina, Domingo Sancho-Knapik, Enrique Arrechea, J. Julio Camarero

**Affiliations:** 1Estación Experimental Aula Dei (EEAD-CSIC), 50059 Zaragoza, Spain; gilpelegrin@eead.csic.es; 2Centro de Investigación y Tecnología Agroalimentaria de Aragón (CITA), 50059 Zaragoza, Spain; jjpeguero@cita-aragon.es (J.J.P.-P.); dsancho@cita-aragon.es (D.S.-K.); 3Instituto Agroalimentario de Aragón-IA2, Centro de Investigación y Tecnología Agroalimentaria de Aragón (CITA), Universidad de Zaragoza, 50013 Zaragoza, Spain; 4Unidad de Suelos y Riegos, Centro de Investigación y Tecnología Agroalimentaria de Aragón (CITA), Estación Experimental Aula Dei (EEAD-CSIC), 50013 Zaragoza, Spain; 5Servicio de Planificación y Gestión Forestal, Gobierno de Aragón, 50071 Zaragoza, Spain; earrechea@aragon.es; 6Instituto Pirenaico de Ecología (IPE-CSIC), 50059 Zaragoza, Spain

**Keywords:** *Abies alba*, atmospheric drought, forest die-off, silver fir, vapor pressure deficit, xylem embolism

## Abstract

Agricultural and ecological systems are threatened by extreme and compound climate extremes such as hotter droughts. These events are characterized by elevated maximum temperatures, leading to atmospheric drought, and reduced precipitation, leading to soil drought. Such conditions reduce plant productivity and are increasing mortality trees in forests worldwide. Some forest types are particularly vulnerable to hotter droughts such as some European mountain silver fir (*Abies alba*) forests. However, we still lack conceptual frameworks linking hotter droughts and rising VPD with growth decline and tree death. This review elucidates physiological responses to drought in conifers with a focus on silver fir. In silver fir declining populations, prolonged stomatal closure under elevated VPD can lead to reduced growth, and impaired xylem development, potentially triggering positive feedback that exacerbates hydraulic limitations. We also review the ecological significance of xylem vulnerability to embolism, identifying the critical water potential thresholds that determine silver fir survival and hydraulic failure risk under soil water deficit. These findings underscore the importance of both atmospheric and soil drought as physiological stressors causing forest decline, and highlight the need for further research into adaptive strategies and early warning indicators in tree species.

## 1. Introduction: Silver Fir Decline in a Global Context

Forests represent a major global carbon reservoir, yet abrupt mortality events can trigger rapid and substantial carbon emissions [1]. Recent widespread reports of forest decline have been attributed to drought stress intensified by climate warming [2]. In arid regions, warming has been linked to rapid canopy loss and selective overstory mortality [3], but the drivers of decline in temperate coniferous forests remain comparatively understudied [4]. The complexity of forest decline arises from multiple interacting stressors operating across spatial and temporal scales, complicating causal inference [5]. Many studies adopt Manion’s framework [6], which distinguishes predisposing (e.g., site conditions), inciting (e.g., drought), and contributing (e.g., biotic agents) stressors that collectively reduce tree vigor [7,8]. Historical land use, such as logging, may predispose forests to decline, yet its role in temperate conifer systems is often overlooked despite widespread past exploitation [9].

The global decline of fir species (*Abies* spp.) has become increasingly evident in recent decades, driven by a combination of climatic stressors, biotic agents, and historical land-use legacies. Silver fir (*Abies alba* Mill.) decline has been widely documented since the 1970s [10], particularly in southern and central Europe, where warming temperatures and recurrent droughts have intensified physiological stress, especially in mature individuals [11,12,13]. These patterns suggest that recent increases in drought stress, potentially driven by warming and altered precipitation regimes, may be contributing to ongoing silver fir decline. Similar patterns have been observed in other fir species such as *Abies chensiensis*, a rare and endangered species from East Asia, which is projected to experience significant range contraction under future climate scenarios, with suitable habitats shifting toward higher elevations [14]. In North America, ecotonal shifts between *Abies concolor* and *Abies magnifica* in the Sierra Nevada of California (USA) suggest that warming is altering regeneration dynamics and competitive hierarchies [15].

Despite its historical classification as drought-tolerant, silver fir has shown increasing vulnerability to water deficit and heat stress, particularly in low-elevation or marginal populations [16]. However, some silver fir stands, whether of anthropogenic or natural origin have shown an excellent response in terms of tree growth and survival and regeneration capacity under mediterranean climatic conditions [17]. In fact, reconstruction and vegetation modeling studies suggest that silver fir may have historically occupied lowland and even coastal areas under warmer-than-present climatic conditions [18,19,20]. Notably, Henne et al. [18] provide palynological evidence indicating the coexistence of silver fir with *Quercus ilex* in forest assemblages that developed near the Mediterranean coast during the warmest phases of the Holocene. A particularly bold aspect of these interpretations is the hypothesis that the disappearance of silver fir around 6000 years before present was not primarily driven by climatic shifts, but rather by severe landscape disturbances such as fire and grazing. According to their models, such vegetation configurations could still be viable under current climatic conditions, provided that temperature increases remain moderate. Importantly, the potential coexistence of *Q. ilex* and *A. alba* is only predicted under low-disturbance simulation scenarios. The interaction between past management practices—such as intensive logging or monastic silviculture—and current climate extremes further complicates the resilience of fir forests [21]. Moreover, human intervention may be responsible for the emergence of pests and diseases [22], some of them (e.g., fungi of genus *Heterobasidion*) being proposed as primary—if not exclusive—drivers of silver fir decline [23].

Consequently, understanding the causes of silver fir decline at one of its southernmost distributional and ecological limits in Europe—namely, the Pyrenean mountains—requires a multidisciplinary perspective. Multiple lines of evidence must be considered, as each may offer insights into the underlying mechanisms of the decline. It is likely that no single factor is solely responsible; rather, a combination of interacting natural and anthropogenic drivers that could be exacerbated in the context of current global change may be contributing to the observed patterns of deterioration. One component of current global change is the increase in the severity of extreme climatic events, such as those characterized by elevated maximum temperatures and reduced water availability, i.e., hotter droughts, which threaten socioecological systems including forests [24]. More frequent and severe hotter droughts are stressing forests worldwide leading to higher mortality rates [25], and reducing their productivity and carbon sink capacity [26,27,28]. These observations have emphasized the key role of climatic compound events triggering hotter droughts (e.g., reduced soil moisture, elevated maximum temperature and evapotranspiration rate) in forest dynamics, particularly regarding tree growth and photosynthesis rates [29].

A major driver of such hotter droughts is the increase in vapor pressure deficit (VPD), which enhances atmospheric dryness and amplifies drought stress [29,30]. Hotter droughts and heat waves can reduce stomatal conductance (g_s_) rates leading to a reduction in water loss through transpiration, but at the cost of lower carbon gain [31]. Such reduction in photosynthesis could translate into lower growth and productivity, predisposing to tree death [29]. However, we still lack comprehensive assessments on the role played by VPD on water-gas exchange in forests and tree species showing a high vulnerability to hotter droughts and atmospheric drought such as European mountain forests dominated by silver fir [11,32]. This review aims to present VPD as an additional factor potentially influencing the physiological response of silver fir in areas where the species exhibits the highest rates of decline. A conceptual model is proposed to describe the relationship between stomatal behavior in response to VPD and its cascading effects on the overall functioning of the tree.

## 2. The Effect of Atmospheric Drought on Stomatal and Photosynthesis Rates in Conifers with a Focus on Silver Fir

The direct response of stomata to varying degrees of atmospheric humidity, with some or even complete independence from the edaphic drought experienced by the plant, has been a subject of intense debate since the early 20th century, as summarized by Sheriff [33]. Pioneering studies supported the idea that VPD is one of the primary environmental factors influencing g_s_ and net CO_2_ assimilation (*A*_N_) [34,35,36]). Currently, there is a consensus that high VPD conditions would reduce g_s_ and *A*_N_ [30,37,38,39], although the underlying mechanisms remain under discussion.

One of the earliest studies regarding the effect of increased VPD on photosynthetic activity in different conifer species was conducted by Sandford and Jarvis [40]. These authors reported markedly different responses among species, highlighting the high sensitivity of *Picea sitchensis* compared to the very low sensitivity of *Pinus sylvestris* to VPD in terms of stomatal closure. Consequently, water loss by transpiration increased significantly in *P. sylvestris* as the atmosphere became drier (higher VPD), whereas transpiration in *P. sitchensis* remained fairly constant across the entire VPD range. Conversely, *A*_N_ remained nearly constant in *P. sylvestris*, while it was clearly reduced in *P. sitchensis* due to stomatal closure.

Grieu et al. [41] provided additional data regarding the specific responses of different conifers to changes in VPD, confirming the heterogeneous behavior within this group of forest species. This study indicated that the effect of increased VPD on *A*_N_ seemed to be greater than that expected solely from the reduction in g_s_. Limitations to CO_2_ diffusion in the mesophyll may underlie this phenomenon, as later studied by Peguero-Pina et al. [42,43] when compared the mesophyll conductance to CO_2_ (g_m_) and photosynthetic activity of silver fir and the Spanish fir (*Abies pinsapo*). This additional effect of VPD over g_m_, combined with stomatal closure, could further impair the ability of the plant for CO_2_ assimilation and, ultimately, the stability of photosynthetic machinery.

Differences in the specific response to increased VPD have been observed even within Mediterranean species of genus *Abies*. Thus, Guehl et al. [44], when compared different circum-Mediterranean fir species, reported that all species exhibited notable stomatal sensitivity to atmospheric dryness. Remarkably, they found that stomatal opening only reached 50% in silver fir and 30% in *Abies bornmulleriana* at VPD = 1 kPa (considered optimal for most plant species). These authors interpreted this response as a way to avoid significant water losses in the needles due to the lack of internal drought tolerance mechanisms. Moreover, Guehl et al. [44] also found that silver fir exhibited a certain independence of *A*_N_ from g_s_, which could be explained by outstanding role of g_m_ as the most limiting factor for CO_2_ assimilation in this species (more than 50% according to Peguero-Pina et al. [42]. Otherwise, the greater dependence of *A*_N_ on g_s_ that was found in *Abies maroccana*, currently considered a Spanish fir (*A. pinsapo*) subspecies, could also be explained [42]. Indeed, this comparative study of the physiological aspects related to carbon assimilation in silver fir and Spanish fir revealed that *A*_N_ in the Spanish fir is more limited by g_s_ (around 50%) than by g_m_ (24%).

The different response to VPD at intraspecific level was also studied for silver fir by Guehl and Aussenac [45], who remarkably hypothesized that “the increased mortality of silver fir in Western Europe may be partly due to its high sensitivity to humidity and water stress”, due to the long-term effects of this conservative stomatal response on tree growth. In this study, they compared two populations of silver fir from highly contrasting origins within France, namely Ecouves (300 m a.s.l., NW France) and Nebias (600 m a.s.l., SE France). The climate in the former location is oceanic with frequent rains, high cloud cover and a low risk of summer drought. By contrast, the climate in Nebias, located in the Pyrenenan foothills, shows a strong Mediterranean influence, with pronounced summer droughts. In fact, these authors considered the habitat of the later population within the most climatically limiting for silver fir in France. The Ecouves firs clearly exhibited higher values of *A*_N_ and g_s_ but a greater sensitivity to increasing VPD than the xeric population (Nebias) under favorable conditions in terms of soil water availability. Interestingly, this pattern was reversed when soil water deficit prevailed in the experiment, which could explain the better capacity of Nebias firs to maintain higher photosynthetic activity under unfavorable environmental conditions.

Guicherd [46] also studied two French silver fir populations under contrasting climatic conditions: a mesohygrophilous stand (hereafter MH) in the outer French Alps and a mesoxerophilous stand (hereafter MX) in the inner French Alps, where the fir grows near its ecological limits. As expected, VPD at midday during summer was clearly higher in MX, reaching values higher than 3 kPa, whereas it was ca. 1.5 kPa in MH. Regarding g_s_, MH displayed values even three times higher than those found in MX during summer. However, the stomatal response to VPD was very similar in both populations, with the stomatal closure occurring at values as low as 0.3 kPa. The author concluded that this is “consistent with their well-known sensitivity to atmospheric humidity”. Despite this, the response patterns of the two populations seemed to exhibit different behaviors. Specifically, while the MH population showed a steep decline in g_s_, the VPD-induced stomatal closure was more gradual in the MX population. Could this behavior be interpreted as indicative of phenotypic or genetic adaptation to more xeric conditions? Although Guicherd [46] already raised this question, he concluded that there is no reason to consider MX a distinct ecotype. Nevertheless, the question remains unresolved.

Taken together, these studies conducted in the late 20th century seem to confirm that silver fir, like other closely related fir species, exhibits a highly sensitive stomatal response to increases in VPD. It should be noted that this phenomenon occurs at VPD values (ca. 1–1.5 kPa) much lower than those recorded for Mediterranean [47] or even transitional Mediterranean habitats [48]. In this regard, Peguero-Pina et al. [49] evidenced that VPD rarely exceeded 1 kPa during summer in a silver fir population in the Spanish Pyrenees, situating this parameter within the previously published range for the species. In a previous study, Peguero-Pina et al. [50] compared the physiological performance of two silver fir populations in the Spanish Pyrenees (Ansó Valley, Huesca) under contrasting climatic conditions (temperature and relative humidity): the wet Gamueta (hereafter site G) site, representing a well-preserved, old-growth and vigorous fir forest, and the dry Paco Ezpela (hereafter site E), with a high proportion of dead or declining individuals showing pronounced canopy defoliation and dieback. We have taken advantage of this dataset and calculated VPD for both populations. The results obtained evidenced marked differences between both sites (Figure 1), with VPD values for site G in the same range than other European silver fir populations (e.g., [51]). By contrast, VPD values for site E were much higher than for site G (Figure 1) and close to the values that can be found in transitional Mediterranean habitats. Assuming a similar response to VPD (which was not investigated by these authors), the higher VPD values in the more xeric site (E) may have negative consequences in the photosynthetic activity of firs from this population. Effectively, *A*_N_ and g_s_ were significantly lower in current-year needles for site E (Figure 2), which clearly reflect a reduced vigor and growth capacity in this declining population.

## 3. Impacts of Atmospheric Drought on Photosystem II (PSII) Performance in Silver Fir

Prolonged stomatal closure over extended periods due to atmospheric drought, while reducing transpiration, may lead to functional disruptions in photosystem II (PSII). Thus, under high VPD, the reduction in g_s_ is the main cause of the decline in *A*_N_ due to disruptions in both photochemical and even biochemical mechanisms involved in photosynthesis [52]. Under these conditions, the complexes responsible for light harvesting continue to function, which implies that there is an excess of light energy that cannot be directed to the electron transport chain for carbon fixation [53]. The presence of electrons not used in CO_2_ fixation can react with O_2_, generating oxidative damage that affects the photosynthetic machinery [54]. To diminish this excess of excitation energy, the downregulation of the maximum potential PSII efficiency at predawn (*F*_V_/*F*_M_) (which is known as photoinhibition) is a common response that can become chronic under extreme stress conditions. This phenomenon was reported by Peguero-Pina et al. [50] for the declining site E (Figure 3) and documented by Kitao et al. [55] and Je et al. [56] in several fir species under conditions impairing carbon fixation. This drop in *F*_V_/*F*_M_ values in site E was accompanied by other signs of alteration of PSII activity, such as the decrease in actual PSII efficiency (Φ_PSII_, Figure 3) and the increase in the de-epoxidation state of the xantophyll cycle pigments ((A + Z)/(V + A + Z), Figure 4) with respect to the values measured in site G. Consequently, needles from site E mostly exhibited significantly lower values of physiological reflectance index (PRI) than those from site G (Figure 5). In this regard, it is worth mentioning that the effect of VPD over different chlorophyll fluorescence parameters has been recently explored by several authors [57,58].

## 4. Xylem Vulnerability to Embolism Under Soil Water Deficit: What Are the Critical Water Potential Values?

When xylem water potential reaches specific thresholds—more likely under conditions of soil water deficit—cavitation may occur due to hydraulic tension, ultimately leading to embolism within the xylem conduits. An embolized conduit is one that becomes filled with air at atmospheric pressure. While this phenomenon is reversible in a few plant species, in most cases embolism results in the permanent loss of xylem hydraulic conductivity. This can compromise plant survival if the affected proportion of the vascular system exceeds specific thresholds—commonly accepted as loss of 50% (P_50_) and 88% (P_88_) of hydraulic conductivity for conifers and angiosperms, respectively [59]. The ability to resist embolism due to water stress is among the most thoroughly studied adaptive traits in forest ecology as an ultimate causing factor of forest dieback [60,61].

The first P_50_ data on silver fir were provided by Cochard [62], who reported results for several conifer species, including *Cedrus atlantica*, *Cedrus deodara*, *Pinus sylvestris*, *Picea abies*, *Pseudotsuga menziesii* and *A. bornmuelleriana*. Specifically, this author determined that P_50_ value for *A. alba* was just above −3.5 MPa. For comparison, P_50_ values for *P. sylvestris and P. abies*—species that often co-occurs with silver fir in mixed stands—were slightly different (ca. −3.0 and −3.6 MPa, respectively). The general trend across species is relatively uniform, with *C. atlantica* standing out for its greater resistance, showing a P_50_ value close to −5 MPa. Therefore, the P_50_ value observed in silver fir is not particularly low, which seems to indicate that this species cannot be considered highly vulnerable to drought-induced cavitation, especially when compared with other co-occurring species. Thus, as previously noted, it is higher (less negative) than that of Scots pine and ca. 1 MPa higher than that of European beech (*Fagus sylvatica*) [63], a common co-occurring species in European forests. In addition, Peguero-Pina et al. [49] compared cavitation vulnerability curves for three silver fir populations and two *Abies pinsapo* populations. These authors did not find significant differences in P_50_, either between species or among provenances. The mean P_50_ value reported for silver fir was −3.80 ± 0.03 MPa (mean ± s.e.), closely aligning with Cochard’s [62] earlier results.

Nourtier et al. [64] investigated the resistance to drought-induced cavitation in silver fir on the Mont Ventoux (southern France), a limestone mountain within a Mediterranean climate zone. Here, silver fir colonized the montane belt over former reforested pine stands. The site is characterized by dry summers (mean summer precipitation over the last 50 years: 100–200 mm at the mountain base; mean summer temperature: 20–25 °C). Motivated by widespread fir mortality in the region, the study reported a surprisingly low P_50_ value of −4.85 MPa, indicating unusually high cavitation resistance in this population. Interestingly, even during the most intense summer droughts, the authors recorded only slight reductions in hydraulic conductivity (around 14%), with midday water potentials rarely dropping below −2 MPa. Similarly, moderate water potential values were recorded by Konôpková et al. [51] in a study of a fir forest in Slovenia during the summer of 2015.

## 5. A Conceptual Model of Silver Fir Decline Based on High VPD and Soil Water Deficit

Overall, some conclusions can be derived from the results reported by Peguero-Pina et al. [50]. First, declining silver fir populations, besides receiving or not a lower amount of annual or summer precipitation, can be affected by a higher atmospheric dryness (higher VPD values) inducing stomatal closure (Figure 6). This mechanism can reduce overall transpiration, keeping plant water potential far from critical cavitation inducing values. In fact, no differences in predawn water potential were found between sites E and G. However, although this mechanism helps trees to avoid risky water potential values and slows soil water depletion, it also limits the ability for carbon assimilation (Figure 6). A high degree of stomatal sensitivity to VPD can be considered as a characteristic of the so-called isohydric (“water-saver”) strategy, quite common among conifers, and could preserve from “death by drought”. However, overly strict regulation of gas exchange to limit water loss can lead to “death by starvation” [3,65] (Figure 6). Under these conditions of carbon starvation, trees experience negative carbon balance which can be translated into very low non-structural carbohydrate (NSC) concentrations [3,66,67,68]. Nevertheless, both mechanisms may be dependent and related because phloem impairment can limit carbon mobilization and NSCs are often used as osmolytes or to build defenses against pathogens [66,69]. Current evidence shows strong support to the hydraulic failure mechanism [70,71], albeit some studies reported low NSC reserves during mortality episodes suggesting positive feedbacks between both mechanisms [72]. More field data are required to disentangle the relative contributions of hydraulic failure and carbon starvation to the dieback of temperate conifers such as fir species, but current evidence has not found severe drops in sapwood NSC during dieback episodes [73]. Empirical approaches could further refine these findings given that experiments have revealed decreases in NSCs in root [74], an organ which is a major carbon sink and hydraulically sensitive in trees.

Even if the decline in carbohydrate assimilation does not reach levels approaching starvation, a lower photosynthetic capacity during summer could jeopardize the growth of the annual rings and reduce the xylem conductive area in these fir stands under dry atmospheres [75]. A progressive reduction in the xylem conductive area over an extended period (years) could lead to a decreased ability of the trees to supply water to the needles, further exacerbating the water stress situation (Figure 6). In this regard, Camarero et al. [73], when studied different conifer species including silver fir, indicated the existence of early warning signals that predict a decline trend. Specifically, declining silver firs already exhibited reduced growth compared to unaffected individuals one to three decades before symptoms became visibly evident. It raises the question of whether the progressive decrease in the conductive xylem area ultimately results in specimens with severe hydraulic problems. This phenomenon is also described by Caillaret al. [76], who detect a global decrease in radial growth prior to mortality in approximately 84% of mortality events, especially among gymnosperms. These species typically show long-term, gradual reductions in growth before dying. Gazol et al. [77] measured the growth of ca. 1300 silver fir specimens across 111 European locations, and they found that there was a reduction in growth for many silver fir individuals in southwestern Europe. Crespo-Antia et al. [12] also found that declining silver firs in two populations of the Spanish Pyrenees showed decreasing growth trends years before manifesting growth decline and crown dieback, confirming that reduced growth rates are reliable predictors of tree vigor.

Ultimately, positive feedback would occur between the decline in *A*_N_ under dry conditions and the development of a conductive system aligned with the functional architecture of the tree (Figure 6). Assuming that xylem rings of silver fir maintain their functionality over several years, hydraulic conductivity will depend on the sum of the conductivities of each annually produced growth ring until reaching its natural operational limit. If the production of rings decreases, particularly over multiple consecutive years, hydraulic conductivity of trunk and branches may progressively decline. This reduction will induce a decreased productive capacity of the plant [78], which will further limit, through the aforementioned feedback, the formation of new conductive surface area (Figure 6).

## 6. Conclusions

Silver fir forests, particularly those located in transitional and Mediterranean-influenced mountain regions, are increasingly vulnerable to the compound effects of atmospheric and soil drought. Elevated VPD emerges as a critical driver of physiological stress, triggering stomatal closure and reducing carbon assimilation, which in turn compromises growth and xylem development. This review highlights the dual threat posed by hydraulic failure and carbon starvation, both of which can initiate a feedback loop that accelerates tree decline and mortality. The possible sensitivity of silver fir to VPD and its conservative water-use strategy (isohydric behavior) may offer short-term protection against cavitation but at the cost of long-term growth and resilience. Evidence suggests that declining individuals often show reduced photosynthetic performance, impaired PSII activity, and diminished xylem conductivity years before visible symptoms appear. These physiological signals can serve as early warning indicators of forest health deterioration. Understanding the thresholds of xylem vulnerability and the interplay between atmospheric and edaphic drought is essential for predicting forest responses to climate change.

Future research should explore the ecophysiology of silver fir in its Pyrenean populations, as current evidence of functional alterations associated with forest decline remains limited. The hypothesis that elevated VPD may trigger a highly conservative water-use response—characterized by stomatal closure—yet detrimental to long-term carbon assimilation, requires more robust empirical validation. The presence of silver fir populations in climates resembling Mediterranean conditions suggests that responses to atmospheric drought may vary within the species. Moreover, *A. pinsapo*, a closely related species with a natural or reforested distribution in Mediterranean environments, may exhibit greater tolerance to high VPD. If confirmed, comparative functional studies between both species could provide valuable insights into the mechanisms underlying VPD responses in genetically related conifers. Finally, studies on the hydraulic functioning of declining individuals—particularly those examining feedbacks between physiological status, growth, and water transport capacity to the canopy—could help validate the conceptual model proposed in this work.

## Figures and Tables

**Figure 1 plants-14-03308-f001:**
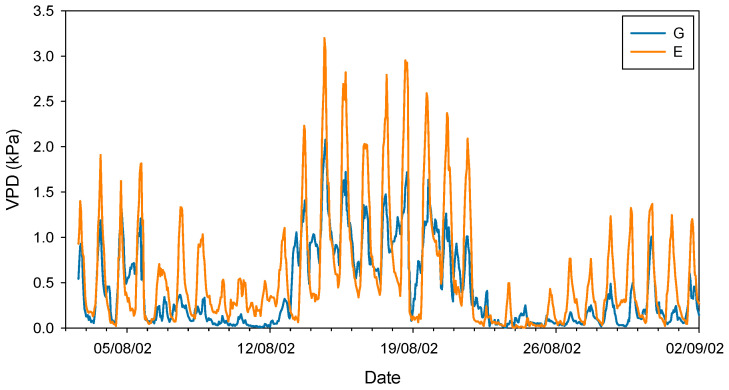
Vapor pressure deficit (VPD) during August 2002 in sites G (Gamueta, blue lines) and E (Paco Ezpela, orange lines) of the Spanish Pyrenees calculated by Peguero-Pina et al. [50].

**Figure 2 plants-14-03308-f002:**
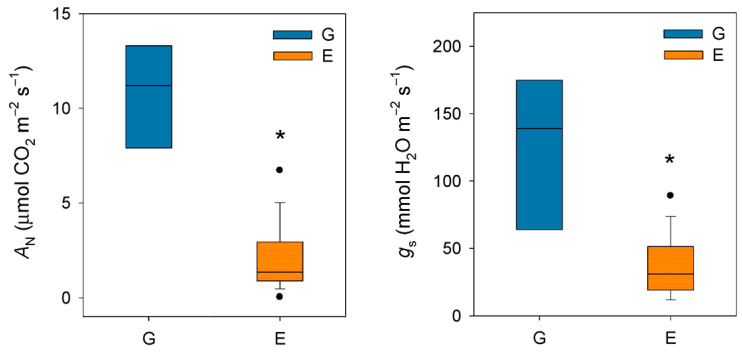
Box plots showing net CO_2_ assimilation (A_N_) and stomatal conductance (g_s_) in current-year needles from silver fir populations located in the Spanish Pyrenees: sites G (Gamueta, blue bars) and E (Paco Ezpela, orange bars). Data are means ± s.e. Asterisks denote significant differences between sites at *p* < 0.05 and points show outliers. Redrawn from Peguero-Pina et al. [50].

**Figure 3 plants-14-03308-f003:**
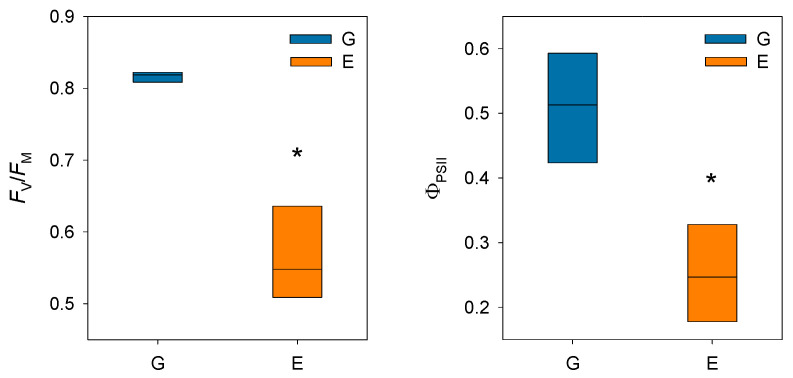
Box plots showing maximum potential PSII efficiency at predawn (*F*_V_/*F*_M_) and actual PSII efficiency (Φ_PSII_) in current-year needles from silver fir populations located in the Spanish Pyrenees: sites G (Gamueta, blue bars) and E (Paco Ezpela, orange bars). Asterisks denote significant differences between sites at *p* < 0.05. Redrawn from Peguero-Pina et al. [50].

**Figure 4 plants-14-03308-f004:**
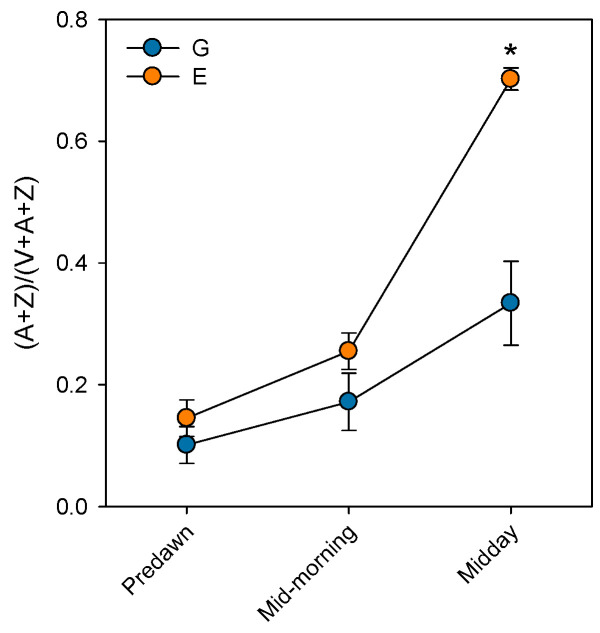
De-epoxidation state of the xantophyll cycle pigments ((A + Z) / (V + A + Z)) measured at predawn, mid-morning (8 h.s.) and midday (12 h.s.) in current-year needles from silver fir populations located in the Spanish Pyrenees: sites G (Gamueta, blue circles) and E (Paco Ezpela, orange circles). Data are means ± s.e. Asterisks denote significant differences between sites at *p* < 0.05. Redrawn from Peguero-Pina et al. [50].

**Figure 5 plants-14-03308-f005:**
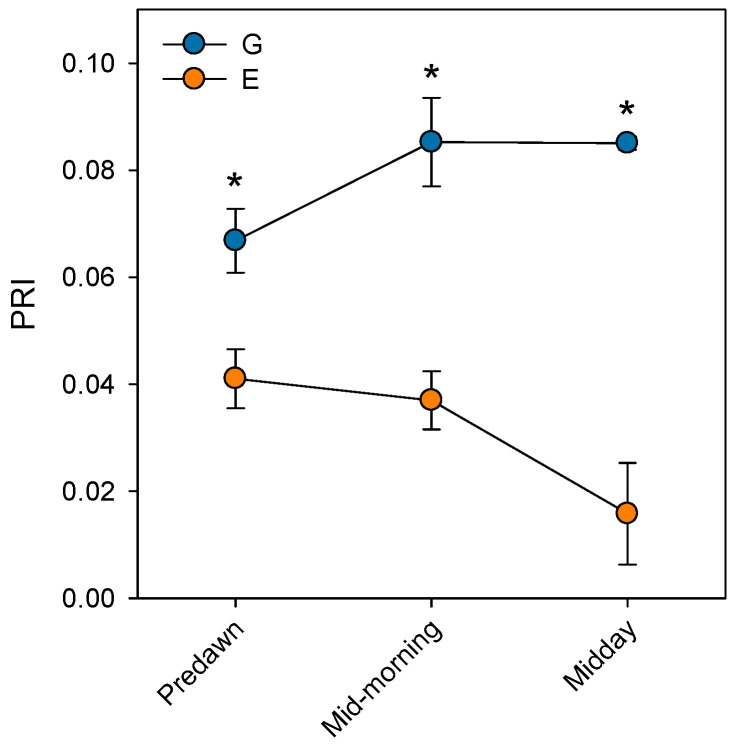
Physiological reflectance index (PRI) measured at predawn, mid-morning (8 h.s.) and midday (12 h.s.) in current-year needles from silver fir populations located in the Spanish Pyrenees: sites G (Gamueta, blue circles) and E (Paco Ezpela, orange circles). Data are means ± s.e. Asterisks denote significant differences between sites at *p* < 0.05. Redrawn from Peguero-Pina et al. [50].

**Figure 6 plants-14-03308-f006:**
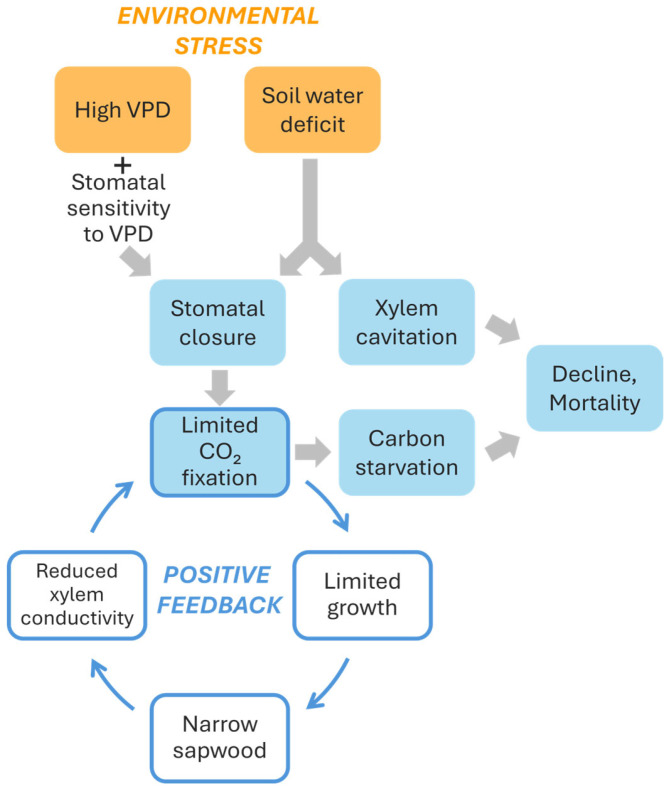
Scheme of the physiological effects of two environmental stresses: high vapor pressure deficit (VPD) and soil water deficit, and their consequences regarding tree dieback and mortality. Notice the positive feedback generated by the limited carbon fixation and limited growth.

## Data Availability

Data are available on reasonable request.

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
