# Peer review of "How Warmer and Drier Conditions Drive Forest Dieback and Tree Death: A Review and Conceptual Model for Silver Fir"

_plants, 2025, doi:10.3390/plants14213308_

Round 1

Reviewer 1 Report

Comments and Suggestions for Authors

The manuscript by Gil-Pelegrín et al. synthesizes the photosynthetic and hydraulic traits of silver fir to explore how atmospheric drought impacts photosynthesis (source activity) and its subsequent effects on growth (sink strength) and mortality risk, proposing a conceptual model of these processes. This work is valuable for understanding the physio-ecological mechanisms in this species. However, the conceptual model presented appears broadly applicable to conifers in general, lacking specificity to silver fir, which limits its novelty. The general pathway—whereby atmospheric and soil drought affect photosynthetic capacity and hydraulic function, leading to reduced radial growth or mortality via carbon starvation and/or hydraulic failure—is well-established in the literature. The proposed model, therefore, reflects a consensus view rather than addressing a highly contested or unclear process. The more pressing challenge in the field lies in quantifying these processes and their relative contributions. For instance, how does carbon starvation and hydraulic failure interactively impact growth? How are they traded off, and do critical thresholds exist? A discussion incorporating non-structural carbohydrates (NSCs) as a key linking factor would significantly strengthen the manuscript in this context. Furthermore, the exclusive focus on a single species, resulting in a review of only 42 articles, limits the review's scope and impact. Expanding the scope to summarize relevant findings on conifers globally—comparing results from similar or different climates with those for silver fir—would make the review relevant to a much broader audience.

Specific Comments:

  • Line 35: I recommend adding a dedicated 'Introduction' section at the beginning to provide broader context and significance of the research area, making the manuscript more accessible to a general readership. The current first section contains some background elements but should be expanded and formally titled 'Introduction'.
  • Line 218: Please check and standardize the citation format here.
  • Line 235: This paragraph feels underdeveloped. The authors list P50 values for the species but do not delve into the relationship between these values and environmental factors. As a reader, I am more interested in the physiological implications of these P50 values. Furthermore, how does this key hydraulic trait specifically contribute to the processes of dieback or mortality mentioned in the title? This crucial link needs elaboration.
  • Line 250 (and similar instances): Several statements in the manuscript are presented as inferences or suggestions. Claims of this nature should be supported by appropriate citations to the primary literature.

Author Response

Reviewer 1 comments

The manuscript by Gil-Pelegrín et al. synthesizes the photosynthetic and hydraulic traits of silver fir to explore how atmospheric drought impacts photosynthesis (source activity) and its subsequent effects on growth (sink strength) and mortality risk, proposing a conceptual model of these processes. This work is valuable for understanding the physio-ecological mechanisms in this species. However, the conceptual model presented appears broadly applicable to conifers in general, lacking specificity to silver fir, which limits its novelty. The general pathway—whereby atmospheric and soil drought affect photosynthetic capacity and hydraulic function, leading to reduced radial growth or mortality via carbon starvation and/or hydraulic failure—is well-established in the literature. The proposed model, therefore, reflects a consensus view rather than addressing a highly contested or unclear process. The more pressing challenge in the field lies in quantifying these processes and their relative contributions.

> The conceptual model presented in this manuscript builds upon previous studies on Abies alba, which suggest that this species is particularly sensitive to vapor pressure deficit (VPD). While the model might appear applicable to conifers in general, we consider such a generalization to be a strong assumption, as there is no clear evidence that all conifers respond similarly to VPD. For instance, Sandford and Jarvis (1986) already showed that Pinus sylvestris exhibits a low stomatal sensitivity to VPD so our model may not be fully applicable to this conifer species. Therefore, as the review focuses on A. alba, we do not state in the manuscript that the model is universally applicable to all conifers. However, we believe that future studies on this topic may refer to our model based on A. alba as a conceptual foundation for their research.

Sandford, A.P.; Jarvis, P.G. Stomatal responses to humidity in selected conifers. Tree Physiol. 1986, 2, 89–103. https://doi.org/10.1093/treephys/2.1-2-3.89

For instance, how does carbon starvation and hydraulic failure interactively impact growth? How are they traded off, and do critical thresholds exist? A discussion incorporating non-structural carbohydrates (NSCs) as a key linking factor would significantly strengthen the manuscript in this context.

> We thank Reviewer 1 for this valuable suggestion. Accordingly, we have revised the manuscript to include a discussion on the interplay between carbon starvation and hydraulic failure, and their respective impacts on plant growth. We also elaborate on the role of non-structural carbohydrates (NSCs) as a crucial linking factor. Please refer to lines 366–379 in the revised version of the manuscript.

Furthermore, the exclusive focus on a single species, resulting in a review of only 42 articles, limits the review's scope and impact. Expanding the scope to summarize relevant findings on conifers globally—comparing results from similar or different climates with those for silver fir—would make the review relevant to a much broader audience.

> We understand that expanding the scope of this review to all conifers would make it relevant to a much broader audience. Nonetheless, the aim of this review was precisely to focus on Abies alba, a European conifer that is well documented to be particularly affected by forest decline. Numerous studies have reported widespread dieback in Abies alba, especially in southern and central Europe, where warming temperatures and recurrent droughts have intensified physiological stress.

Nevertheless, the new Introduction section has been written taking into account this suggestion of including other conifers. Thus, we mention  other fir species, such as Abies chensiensis in East Asia or Abies concolor and Abies magnifica in North America (lines 57 to 62).

Line 35: I recommend adding a dedicated 'Introduction' section at the beginning to provide broader context and significance of the research area, making the manuscript more accessible to a general readership. The current first section contains some background elements but should be expanded and formally titled 'Introduction'.

> In response to this suggestion, we have expanded the background section, which is now formally titled Introduction. In this section, we begin by introducing the phenomenon of forest decline and the main stress factors involved. Next, we examine the decline of Abies alba and other species within the Abies genus. We also highlight key historical aspects of A. alba's distribution. Finally, we propose atmospheric vapor pressure deficit (VPD) as a potentially important factor contributing to fir decline and describe the objective of the review.

Line 218: Please check and standardize the citation format here.

> The citation has been checked and corrected. Thank you.

Line 235: This paragraph feels underdeveloped. The authors list P50 values for the species but do not delve into the relationship between these values and environmental factors. As a reader, I am more interested in the physiological implications of these P50 values. Furthermore, how does this key hydraulic trait specifically contribute to the processes of dieback or mortality mentioned in the title? This crucial link needs elaboration.

> The main aim of this paragraph is to highlight the importance of the ability to resist embolism due to water stress as an adaptive trait in forest ecology and forest dieback. In this regard, we have compared P50 values of silver fir reported in the literature with other conifers and/or other co-occurring species. We conclude that the P50 value observed in silver fir is not particularly low, which seems to indicate that this species cannot be considered highly vulnerable to drought-induced cavitation, especially when compared with other co-occurring species. This is now better explained in this section (line 314 and lines 322-325).

Line 250 (and similar instances): Several statements in the manuscript are presented as inferences or suggestions. Claims of this nature should be supported by appropriate citations to the primary literature.

> Thanks for the suggestion. We have added a significant number of new references to support the statements here presented.

Reviewer 2 Report

Comments and Suggestions for Authors

Report on "How Warmer and Drier Conditions Drive Forest Dieback and Tree Death: A Review and Conceptual Model for Silver Fir".

This manuscript review discusses the ecophysiological responses and risks of silver fir to warming and drying associated with climate change. Temperature increases and drying associated with climate change are progressing seriously, and elucidating the responses of various plant species in forest ecosystems to these changes is a globally important research topic. Silver fir occupies an important position in European forest ecosystems, and knowledge accumulated there about the responses of stomata and photosynthesis to high temperatures and drought could provide extremely useful information for tree species in other regions of the world. I felt that this review report comprehensively summarized the findings of research on silver fir.

Some Comments:
1: The authors go into more detail from the opening section of "The increasing relevance of atmospheric water demand to forests." While I prefer openings that start with in-depth content like this, I think it would be easier for readers to grasp the overall picture if the author provided a brief introduction that provided a broader overview of the manuscript and explained the story behind the argument. Essentially, I think it would be good to have a section similar to the "objectives" section in a typical Original Article.

2: Also, related to the above comment, I think the conclusion section well summarizes the findings from previous research. However, it seems a little unclear what needs to be further focused on and clarified, such as future research themes that could be considered based on current knowledge, or problems that need to be overcome. Each section in the present Manuscript has shown some questions, but it would be good to present the authors' research team's vision for key future challenges.

Author Response

Reviewer 2 comments

1: The authors go into more detail from the opening section of "The increasing relevance of atmospheric water demand to forests." While I prefer openings that start with in-depth content like this, I think it would be easier for readers to grasp the overall picture if the author provided a brief introduction that provided a broader overview of the manuscript and explained the story behind the argument. Essentially, I think it would be good to have a section similar to the "objectives" section in a typical Original Article.

> In response to this suggestion and the one made by Rev. 1, we have provided a brief introduction section, which is now formally titled Introduction, similar to the "objectives" section in a typical Original Article. In this new Introduction, we begin by launching the phenomenon of forest decline and the main stress factors involved. Next, we examine the decline of Abies alba and other species within the Abies genus. We also highlight key historical aspects of A. alba's distribution. Finally, we propose atmospheric vapor pressure deficit (VPD) as a potentially important factor contributing to fir decline and describe the objective of the review.

2: Also, related to the above comment, I think the conclusion section well summarizes the findings from previous research. However, it seems a little unclear what needs to be further focused on and clarified, such as future research themes that could be considered based on current knowledge, or problems that need to be overcome. Each section in the present Manuscript has shown some questions, but it would be good to present the authors' research team's vision for key future challenges.

> We have now added several sentences at the end of the conclusion section (lines 442 to 455) that clearly outline directions for future research. These additions articulate our perspective on the most pressing challenges ahead, emphasizing the need for deeper ecophysiological investigations, comparative species analyses, and the integration of advanced remote sensing technologies. By presenting a structured vision of the next steps, we aim to guide subsequent studies and foster a more comprehensive understanding of the mechanisms driving forest decline under increasing atmospheric drought stress.

Reviewer 3 Report

Comments and Suggestions for Authors

The review summarizes contemporary knowledge on physiology responses in conifers, particularly in Abies alba, to drought in the face of present and predicted climate changes. The Authors analyse results from physiological studies concerning stomatal activity and photosynthesis, photosystem II performance and xylem vulnerability to embolism and next they introduce a conceptual model of silver fir decline based on high VPD and soil water deficit.

The paper is interesting, Authors’ statements are convincing. It is an attempt to summarize the Authors’ own long-term research on this topic, thus it is natural that the Authors include numerous own citations. In my opinion this is justified, the Authors’ have expanded our knowledge in this area in their previous studies. The quality of the manuscript is good. The manuscript is appropriate to be published in 'Plants' but there are some points which might be improved:

Line 41: apart from Ref 3, some more appropriate references would be valuable;

Line 55: I encourage the Authors to include some more new references studying the effect of VPD on photosynthesis rates in this Chapter, e.g. Kurjak et al 2023 (DOI:10.32615/bp.2023.017);

Line 85: please indicate clearly if it is a value of VPD;

Line 90: Ref. No. 12 is not Peguero-Pina et al., please insert the proper number of this reference;

Line 93: the same as given above;

Line 179: I encourage the Authors to include some more references studying the effect of VPD on ChF in this Chapter, e.g. Jin et al. 2022 (DOI:10.1016/j.ecolind.2022.109679), Marozas et al. 2019 (DOI: 10.2480/agrmet.D-18-00004 );

Line 198: I encourage the Authors to broaden the discussion in this chapter and the next one based on some newer papers, e.g. Hayek et al. 2022 (DOI: 10.1111/gcb.16146), Tang et al. 2023 (DOI:10.1111/1365-2435.14408), Mátyás et al. 2021(DOI: 10.3390/f12070821) – usually, the more recent references in a review paper, the strongest its value.

Author Response

Reviewer 3 comments

Comments and Suggestions for Authors

The review summarizes contemporary knowledge on physiology responses in conifers, particularly in Abies alba, to drought in the face of present and predicted climate changes. The Authors analyse results from physiological studies concerning stomatal activity and photosynthesis, photosystem II performance and xylem vulnerability to embolism and next they introduce a conceptual model of silver fir decline based on high VPD and soil water deficit.

The paper is interesting, Authors’ statements are convincing. It is an attempt to summarize the Authors’ own long-term research on this topic, thus it is natural that the Authors include numerous own citations. In my opinion this is justified, the Authors’ have expanded our knowledge in this area in their previous studies. The quality of the manuscript is good. The manuscript is appropriate to be published in 'Plants' but there are some points which might be improved:

Line 41: apart from Ref 3, some more appropriate references would be valuable.

> The introduction section has been rewritten, and more references have been added in this sentence. Please see line 96.

Line 55: I encourage the Authors to include some more new references studying the effect of VPD on photosynthesis rates in this Chapter, e.g. Kurjak et al 2023 (DOI:10.32615/bp.2023.017).

> We have added this new reference in the revised version of the manuscript.

Line 85: please indicate clearly if it is a value of VPD.

> Yes, the value of 1 KPa refers to a VPD value. This is now clarified in the text (line 149).

Line 90: Ref. No. 12 is not Peguero-Pina et al., please insert the proper number of this reference. Line 93: the same as given above.

> We thank Rew. 3 for finding these two mistakes. Effectively, Peguero-Pina et al. was not number 12 in the old version of the manuscript. This reference was number 16, referring to Peguero-Pina et al. 2012, which now (after the incorporation of new references) corresponds to number 42 in the revised version of the manuscript.

Line 179: I encourage the Authors to include some more references studying the effect of VPD on ChF in this Chapter, e.g. Jin et al. 2022 (DOI:10.1016/j.ecolind.2022.109679), Marozas et al. 2019 (DOI: 10.2480/agrmet.D-18-00004 ).

> We have added these two new references in the revised version of the manuscript according to the suggestion of the Reviewer.

Line 198: I encourage the Authors to broaden the discussion in this chapter and the next one based on some newer papers, e.g. Hayek et al. 2022 (DOI: 10.1111/gcb.16146), Tang et al. 2023 (DOI:10.1111/1365-2435.14408), Mátyás et al. 2021(DOI: 10.3390/f12070821) – usually, the more recent references in a review paper, the strongest its value.

> Thank you for these references. They are now incorporated in the text of the revised version of the manuscript.

Round 2

Reviewer 1 Report

Comments and Suggestions for Authors

The authors have fully considered my feedback and made the corresponding revisions. I believe the manuscript is now suitable for publication in this journal.